# Handheld-Load-Specific Jump Training over 8 Weeks Improves Standing Broad Jump Performance in Adolescent Athletes

**DOI:** 10.3390/healthcare10112301

**Published:** 2022-11-17

**Authors:** Wei-Hsun Tai, Hai-Bin Yu, Ray-Hsien Tang, Chen-Fu Huang, Yan Wei, Hsien-Te Peng

**Affiliations:** 1School of Physical Education, Quanzhou Normal University, Quanzhou 362000, China; 2Graduate Institute of Sport Coaching Science, Chinese Culture University, Taipei 11114, Taiwan; 3Department of Physical Education, National Taiwan Normal University, Taipei 106209, Taiwan; 4Department of Physical Education, Chinese Culture University, Taipei 11114, Taiwan

**Keywords:** standing broad jump, 8-week training, impulse, handheld load, sport biomechanics

## Abstract

This study investigated the effects of handheld-load-specific jump training on standing broad jump (SBJ) performance in youth athletes and the biomechanics changes involved. Methods: Fifteen male athletes (mean age, body weight, height, and body mass index were 14.7 ± 0.9 years, 59.3 ± 8.0 kg, 1.73 ± 0.07 m, 19.8 ± 2, respectively) underwent 15 SBJ training sessions over 8 weeks. The data were collected over three phases: before training, after training, and after training with 4 kg loading. Ten infrared high-speed motion-capture cameras and two force platforms, whose sampling rates were 250 and 1000 Hz, respectively, were used to record the kinematic and kinetic data. Visual three-dimensional software was used for the data analyses. Results: Jump performance and all biomechanics variables, including joint and takeoff velocities, ground reaction force, takeoff impulse, and mechanical outputs, improved after training. Conclusions: SBJ training under handheld loading resulted in considerable acute improvements as well as training transfer after 8 weeks. Moreover, explosive ability was effectively enhanced. The present findings serve as a reference for SBJ assessment and jump-related training.

## 1. Introduction

A standing broad jump (SBJ) is often used to assess strength of the lower extremity due to its convenience and efficiency [1,2]. It is also useful for evaluating sprinting ability across various ages as well as for conducting strength and conditioning training among adolescent athletes [3]. The SBJ is also often practiced as plyometric training, that is safer and more effective than conventional weight training, for adolescents to develop explosiveness of the lower extremity [3,4,5]. It benefits strength and explosiveness in leg extensors [6], coordination between leg agonist and antagonist muscles, and motor unit recruitment efficiency of leg muscles [7]. Periodization training with the SBJ enhanced lower-extremity stiffness and further improved running performance [8].

Previous studies demonstrated that SBJ performance was improved with handheld loads by increasing the takeoff distance through forward horizontal body position and center of mass (CoM) [9,10,11], increased takeoff velocity [10,12,13,14], and vertical and horizontal ground reaction force (GRF) and impulse [9]. The lower extremity is intensified via loaded jump training [3,5,9]. Moreover, the training of a SBJ with handheld loads can also enhance neuromuscular ability to improve a sprint performance [10,12,14,15,16].

The SBJ with handheld loads helps to generate a greater ground-reaction force and impulses (vertical and horizontal) during the takeoff that may augment elastic energy of the muscle–tendon complex to boost the generated power amplification [8,12]. Previous researchers indicated that greater handheld loading slows down the muscles’ contraction velocity to benefit greater muscular force production [15]. The SBJ with handheld loads improved neuromuscular mechanisms for stretch shortening cycle (SSC) performance and further changed the elastic behavior of joint muscles sub-components [17].

Previous study suggested that most physical qualities are trainable throughout maturation [18]. However, an incorrect exercise technique or poor supervision increases the resistance training (RT) injury risk of adolescent athletes [19]. However, the benefits of SBJ training under handheld loading have been well established for mature athletes in males [11] and females [14,15]. However, the majority of these studies did not include adolescent athletes. Therefore, as a simple way that is easy to adapt in training, and may be better for developmental stage athletes, using handheld loads during SBJ training could have a number of interesting training implications. The purpose of this study was to explore if an eight-week handheld-load-specific jump training program can improve adolescent athletes’ SBJ performances in biomechanics. We hypothesized that SBJ training with handheld loads would improve adolescent athletes’ explosive performance; also handheld loads may optimize jumping coordination after training.

## 2. Materials and Methods

### 2.1. Participants

Fifteen adolescent male track-and-field athletes who were familiar with the SBJ participated in this study (mean age, body weight, height, and body mass index were 14.7 ± 0.9 years, 59.3 ± 8.0 kg, 1.73 ± 0.07 m, 19.8 ± 2, respectively). None of the participants had sustained lower-extremity injury in the 6 months prior to testing. The study protocol was approved by the university ethics committee, and written informed consent was obtained from all the participants before the commencement of the study.

### 2.2. Procedures

Before the training program began, the participants were instructed to perform SBJ without handheld loads as pre-tests (Pre) in the laboratory and with free arm swinging. The movement sequence was that the jumper swung back and forth to coordinate the following takeoff, and swung forwards at takeoff, then backwards before landing. Since the training program included SBJ with handheld loads, they were asked to complete three SBJs with 1, 2, 3, 4, and 5 kg handheld loads in order to find their optimal handheld loads that could lead to their best SBJ performances [20]. Each participant’s optimal handheld load was determined based on curve fitting for SBJ performance and the average optimal handheld load was 3.8 kg. Optimal SBJ performance was exhibited for total load of 4 kg handheld loads (i.e., each hand holding a 2 kg dumbbell). This load was subsequently used in the 8-week training program in the present study (Table 1). After the 8-week training program, the participants were asked to perform SBJ without and with 4 kg handheld loads as post-tests (Post and PostL, respectively) in the laboratory. Each participant performed three trials of SBJs in Pre, Post and PostL with a 3 min rest between trials. SBJs in Post and PostL were tested in a random order.

### 2.3. Instruments

Kinematic data were collected using ten infra-red high-speed motion-capture cameras (Vicon MX 13+, Oxford Metrics Ltd., Oxford, UK) with a 250-Hz sampling rate, and kinetic data were collected using two force platforms (60-cm × 90-cm, Kistler, Instruments, Inc., Winterthur, Switzerland) with a 1000-Hz sampling rate. The kinematic and kinetic data collection was synchronized by Nexus system (Nexus 1.4, Oxford Metrics Ltd., Oxford, UK). Body segments were defined used a modified Helen Hayes configuration with 69 retroreflective markers (19 mm in diameter) which were placed on the full body of subjects (Figure 1 and Figure 2).

### 2.4. Data Processing

The marker trajectory and force data were identified within the Nexus and Kistler (Bioware 3.2, Kistler, Instruments, Inc., Winterthur, Switzerland) 3.2 software, respectively. The Visual 3D software (C-motion, Rockville, MD, USA) was used to analyze the kinematics and kinetics data after the raw data were exported to a C3D file format and further imported. A low-pass Butterworth digital filter with 6 Hz cut-off frequency was used to decreased random noise during the digitizing process. The joint moment and power were calculated from the kinematic and GRF data [21]. Push-off time was a duration which calculated from the start of downward GRF exceeded 20 N to takeoff [16]. The impulse was calculated from the integration of the GRF–time curve between body weight and takeoff [14]. Figure 3 shows the measurement of the SBJ that separated into takeoff distance, air distance, and landing distance three parts. The total distance of SBJ was defined as the summation of the takeoff distance, air distance, and landing distance [22].

SPSS 20.0 (IBM SPSS Statistics, Somers, NY, USA) was used to perform the statistical calculating. Descriptive statistics (mean, M; standard deviation, SD) were used to determine the characteristics of the participants. The Kolmogorov–Smirnov test was used for normally distributed test. Repeated-measures one-way ANOVAs were applied to all biomechanical variables to determine significance among Pre, Post and PostL SBJs. The Bonferroni post hoc analysis was used to determine pairwise differences. The level of statistical significance was set at 0.05. For the practical relevance of the significant influences, the effect size of the training stage differences was also calculated when the partial η2 between 0.010 and 0.059, between 0.060 and 0.139, and for 0.140 and above indicated small, moderate, and large differences, respectively [23].

## 3. Results

Table 2 shows the kinematic results. The total distance (both *p* < 0.001), takeoff distance, air distance, landing distance, CoM displacement, CoM-H velocity, peak ankle angular velocity, peak hip angular velocity of Post and PostL were significantly greater than those of Pre. The landing distance (*p* < 0.001) and CoM-H velocity of PostL were significantly greater than those of Post.

The kinetic results shown in Table 3, the horizontal and vertical impulse, peak horizontal GRF, peak knee moment, peak power of ankle, and knee of Post and PostL, were significantly greater than those of Pre. (all *p* < 0.001). The push time (*p* < 0.001) and peak horizontal GRF (*p* = 0.002) of PostL were significantly greater than those of Post. The peak hip moment (*p* < 0.001), peak knee and hip power (both *p* < 0.001) of Post were significantly greater than those of PostL.

## 4. Discussion

The main finding of the study was that the training of jumps with handheld loads can significantly improve adolescent athletes’ SBJ performances in biomechanics. Their total jump distance, horizontal takeoff velocity, vertical and horizontal impulse, peak horizontal GRF, peak moment of the ankle, knee and hip, and peak power of the ankle, knee and hip increased by 17.96%, 11.8%, 21.1%, 8.8%, 16.9%, 8.2%, 10.5%, 21.4%, 25.3%, 22.6%, and 15.5%, respectively. It was consistent with the findings of previous studies [3,5,24,25]. As presented in Table 2 and Table 3, with an external force (handheld loads) stimulated, this could increase the adaptability of the muscles, which can be regarded as an overloading effect during training [24,26,27]). In addition, the muscle memory adaptations learned through periodization led to improvements in the SSC performance and jump coordination [28]. The greater GRF [24] and positive impulse [8] contribute to the jumping mechanical output, regardless of takeoff direction (vertical or horizontal). In the present research, the horizontal impulse and GRF were also improved through training. These indicated that the handheld loading in SBJ training may be more advantageous to adolescent athletes to gain greater explosive performance.

Studies have indicated that the arm swing motion is one of the most important influencing factors of SBJ performance [29,30,31,32], especially under loading; the results for jump distance, horizontal takeoff velocity, and impulse were much more affected [9,10,16]. Due to the aforementioned reason, the arm swing movement also benefits the training protocol, because of the favorable neuromuscular adaptations involving the coordination of the arm swing with the movement of the rest of the body, which facilitates the optimization of the jump technique and thus a substantial improvement in SBJ performance in the present study. It is thus quite likely that jumps performed with SBJ provided a substantial horizontal and vertical neuromuscular training stimulus during the concentric phase, which may have stimulated an increase in lower extremity muscle strength [33]. 

Consistent with results of previous studies, the takeoff velocity, push-off time, and peak horizontal GRF were considerably higher in the PostL condition, which indicated the effect of the 4 kg load on jump performance [9,10,11,12,13,14,15,16]. CoM displacement and knee joint velocity were lower with than without loading; however, horizontal jump distance as well as ankle and hip joint velocity were higher without loading than with loading. The aforementioned results confirmed the pull effect of the arm swing motion under loading [31,34,35] and its influence on CoM [9,10,11], which emphasized the higher shoulder torque to pull the trunk toward the direction of the load and increase the distance of CoM [14,29].

The lower extremity mechanical output exhibited improvement after training. Hip moment and power presented interesting results to the similar mechanical outputs in the Pre and PostL conditions but were lower than the Post condition. Compared with Pre, this explains the improvement of SBJ training on lower extremity mechanical output during the 8-week period. However, the smaller mechanical output under the PostL condition may be attributed to the increased pull under loading, which reduced the hip power [36]. Some studies have indicated that reduced joint moment may improve jump performance [37], which may be due to arm swing effects [31,32] and this study found the handheld load reduced joint moment, which was reflected in the PostL condition. Comparing the results of Post and Pre conditions in present research, the higher joint moment in Post may prove that 8 weeks of SBJ training was beneficial for mechanical output. Improvements in lower extremity mechanical ability might be attributed to neuromuscular adaptations such as increased strength and explosiveness of the leg extensor muscles [6], and a longer push-off phase may allow for the coordination of an optimal takeoff position, where slower knee and hip joint movement was beneficial for muscle activation with a load. It is possible that these neuromuscular and strength-explosive adaptations influenced the biomechanical factors related to jump actions, such as joint velocity of the takeoff, which potentially contribute to produce a higher SBJ performance, cumulatively or individually. Overall, the results reflect the benefits of SBJ training under handheld loading.

The significant improvement in SBJ performance in the male adolescent athletes in the present study (*p* < 0.05) may not carry over to athletes of a different age or sex. The development of strength or power is an important issue for adolescent athletes. The present study used a simple method for sport-specific technical training which may decrease injury risk and improve training quality with coaches. Therefore, models of optimal handheld load as well as the short-term (3–6 weeks) and long-term (>8 weeks) effects of similar training programs, with the balance ability alone, or with technical development between bilateral and unilateral under SBJ training are worthy of future exploration. The findings of the present study are highly applicable to trainers and athletes with regard to training program design and exercise prescription. There were some limitations in the present study. The strength of the lower extremity and jumping ability of the youth athletes were not measured prior to testing, and gender was not considered in the present research; these were methodological limitations and an internal problem for maturation.

## 5. Conclusions

SBJ training under handheld loading led to acute improvements in jump performance and training transfer after 8 weeks. Such training programs are useful for explosive exercise training in youth athletes. The findings of this study serve as a reference for SBJ assessment and jump-related training. Most push-off-related variables demonstrated improvements after training, which indicated that the training effectively enhanced the explosive ability of the participants. To refine relevant training programs, future studies can assess whether the landing pattern of SBJ changes under handheld loading. In addition, to optimize training adaptations, SBJ training strategy should be adequately applied in a more complex, such as a unilateral or other, direction and an optimal loading that is suitable for subjects should be tested before training.

## Figures and Tables

**Figure 1 healthcare-10-02301-f001:**
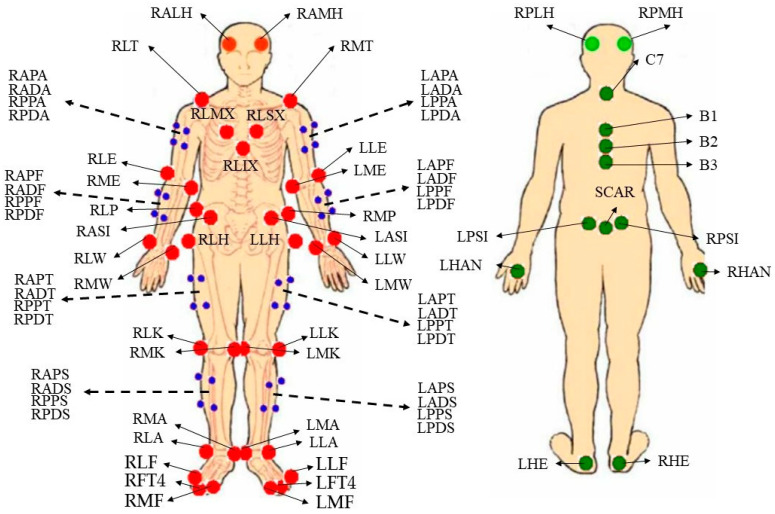
The retroreflective marker locations of modified Helen Hayes configuration.

**Figure 2 healthcare-10-02301-f002:**
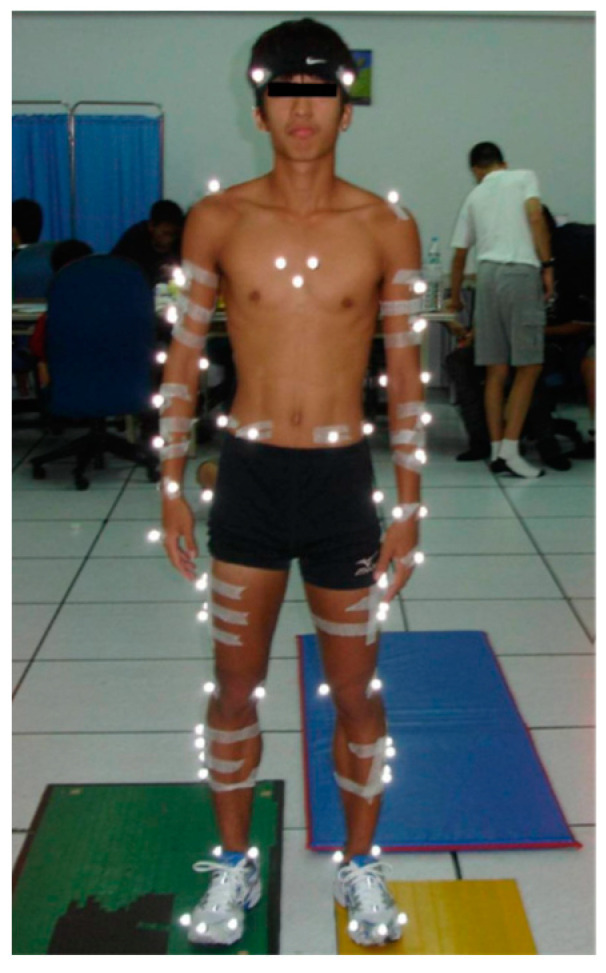
The retroreflective marker locations.

**Figure 3 healthcare-10-02301-f003:**
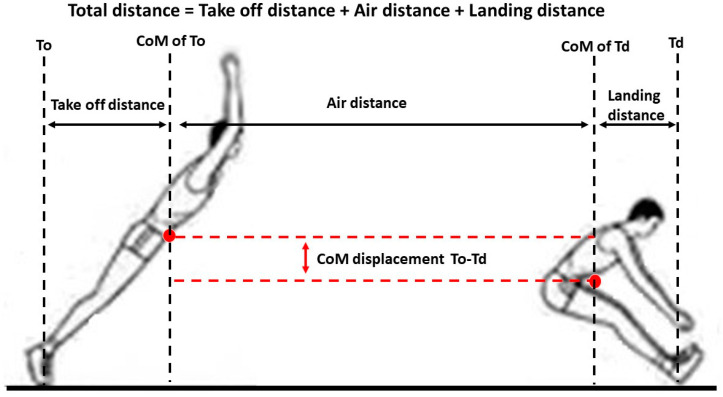
Definitions of standing broad jump. To, takeoff; CoM, center of mass; Td, touch-down.

**Table 1 healthcare-10-02301-t001:** 8 weeks training program (training 100 times per week).

		Sets × Repetitions							
Exercise	Loading (kg)	Week 1	Week 2	Week 3	Week 4	Week 5	Week 6	Week 7	Week 8
VJ	n/a	5 × 10							
ASL	1	5 × 10							
SBJ	n/a		5 × 10						
ASL	2		5 × 10						
VJL	1			5 × 10					
ASL	3			5 × 10					
SBJL	2				5 × 20				
SBJL	2					5 × 10			
ASL	4					5 × 10			
SBJL	3						5 × 10		
ASL	5						5 × 10		
SBJL	4							5 × 10	
ASL	4							5 × 10	
SBJL	4								5 × 10
SBJ	n/a								5 × 10
Total		100	100	100	100	100	100	100	100

n/a = no loading; VJ = Vertical jump; ASL = Arm swing with handhold loads; SBJ = Standing broad jump; VJL = Vertical jump with handhold loads; SBJL = Standing broad jump with handhold loads.

**Table 2 healthcare-10-02301-t002:** Kinematic variables of SBJ performances.

Variables	Pre		Post		PostL		*F*	*η* ^2^	*p*
	M	SD	M	SD	M	SD			
Total Distance (m)	206	20	243 ^a^	19	243 ^a^	19	110.46	0.888	0.000
Takeoff distance (cm)	56	7	60 ^a,c^	7	59	6	10.09	0.419	0.001
Air distance (cm)	125	12	150 ^a,c^	11	137 ^a^	13	33.62	0.706	0.000
Landing distance (cm)	24	5	33 ^a^	4	47 ^a,b^	5	142.00	0.910	0.000
CoM displacement (cm)	24	5	39 ^a,c^	2	28 ^a^	2	102.14	0.879	0.000
CoM-H velocity (m/s)	3.05	0.29	3.41 ^a^	0.23	3.48 ^a,b^	0.19	76.65	0.846	0.000
CoM-V velocity (m/s)	1.84 ^c^	0.21	1.76 ^c^	0.08	1.62	0.06	10.10	0.419	0.004
Peak ankle angular velocity (°/s)	588.7	86.9	745.5 ^a,c^	67.1	714.3 ^a^	66.0	103.05	0.880	0.000
Peak knee angular velocity (°/s)	684.5 ^c^	66.4	700.5	82.4	657.4	86.0	9.07	0.393	0.001
Peak hip angular velocity (°/s)	374.5	44.7	477.9 ^a,c^	37.8	448.2 ^a^	46.4	136.92	0.907	0.000

^a^ significantly greater than Pre; ^b^ significantly greater than Post; ^c^ significantly greater than PostL; *p* < 0.05.

**Table 3 healthcare-10-02301-t003:** Kinetic variables of SBJ performances.

Variables	Pre		Post		PostL		*F*	*η* ^2^	*p*
	M	SD	M	SD	M	SD			
Push-off time (ms)	41	3	42	1	46 ^a,b^	2	30.50	0.685	0.000
H-impulse (N·s)	125.7	19.9	152.1 ^a^	22.9	153.0 ^a^	25.8	36.13	0.721	0.000
V-impulse (N·s)	241.0	37.6	263.7 ^a^	42.0	269.4 ^a^	50.9	10.92	0.438	0.000
Peak-H GRF (N)	597.7	76.2	699.8 ^a^	112.6	723.9 ^a,b^	102.4	32.75	0.701	0.000
Peak-V GRF (N)	1200.4	220.2	1252.4	204.2	1261.8 ^a^	1.96	3.79	0.213	0.035
Peak moment (Nm)									
Ankle	136.7	26.3	147.2 ^a^	26.1	143.4	31	4.87	0.258	0.015
Knee	208.5	52.4	230.5 ^a^	56	245.9 ^a^	57.3	10.87	0.437	0.000
Hip	138.1	19.1	166.8 ^a,c^	16.6	143.3	21.9	42.24	0.751	0.000
Peak power (Watt)									
Ankle	897.2	179.8	1124.1 ^a^	277.6	1056.7 ^a^	201.8	25.88	0.649	0.000
Knee	1613.3	341.7	1991.5 ^a,c^	452.9	1804.4 ^a^	382.5	19.81	0.586	0.000
Hip	1046.1	148.4	1209.7 ^a,c^	177.2	1039.5	142.5	30.63	0.686	0.000

^a^ significantly greater than Pre; ^b^ significantly greater than Post; ^c^ significantly greater than PostL; *p* < 0.05.

## Data Availability

The data presented in this study are openly available in National digital library of theses and dissertations in Taiwan at https://hdl.handle.net/11296/n43syb, accessed on 1 October 2022.

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
