# Peer review of "Handheld-Load-Specific Jump Training over 8 Weeks Improves Standing Broad Jump Performance in Adolescent Athletes"

_healthcare, 2022, doi:10.3390/healthcare10112301_

Round 1

Reviewer 1 Report

This manuscript presents a study of handheld standing-board-jump (SBJ) to investigate if hand-held-load-specific jump training would benefit SBJ performance. The results show that the SBJ performance is improved after training in biomechanical aspects. The manuscript is well written, however, the goal/novelty of this manuscript is unclear to me given how the work is presented and discussed.  

Line 56: “adolescent Although” doesn’t seem to be grammatically correct. 

Lines 56-59: the authors stated that the gap in the current research is not being able to distinguish the contribution from SBJ technique and physical quality. However, the purpose of this study (explore if a jump training program can improve performance) and hypothesis doesn’t seem to relate to this gap (the training could improve performance). They sound very like previous studies the authors mentioned. The authors need to rephrase this paragraph to make it a stronger statement on what’s new/novel about this manuscript. 

Line 96: It would help to have a reference or a diagram of the modified Helen-Hayes configuration for study repeatability 

Lines 153-160: How the subjects were instructed should be in the 2.2 procedure. How previous studies did it should be in Introduction.

The discussion is not very convincing, especially with the unclear statement in the Introduction I mentioned above. By mentioning a lot of consistency with previous studies, it makes me feel even less novelty in this study. If the authors want to quantify the physical quality of the subjects, they should do a muscle strength test. The estimated torques from the experiments don’t tell much, since SBJ is a complicated exercise that requires a series of muscle cooperations. An increase in the joint moment won’t explain or prove muscle strength improvement. The authors also mentioned in lines 175-177 that reduced joint moments may improve jump performance. This further mixed the argument or statement the authors trying to make. Thus, I believe the authors need to spend more time clarifying their statement and distinguishing strength, explosiveness, performance, and technique, and how they should be quantified and related to each other in this study.

Author Response

This manuscript presents a study of handheld standing-board-jump (SBJ) to investigate if hand-held-load-specific jump training would benefit SBJ performance. The results show that the SBJ performance is improved after training in biomechanical aspects. The manuscript is well written, however, the goal/novelty of this manuscript is unclear to me given how the work is presented and discussed.  

Response: We appreciate your kind advice.  The manuscript has been revised

Line 56: “adolescent Although” doesn’t seem to be grammatically correct. 

Response: Thanks for your comments. These have been revised. Please refer to line 56.

Lines 56-59: the authors stated that the gap in the current research is not being able to distinguish the contribution from SBJ technique and physical quality. However, the purpose of this study (explore if a jump training program can improve performance) and hypothesis doesn’t seem to relate to this gap (the training could improve performance). They sound very like previous studies the authors mentioned. The authors need to rephrase this paragraph to make it a stronger statement on what’s new/novel about this manuscript. 

Response: Thanks for your comments. These have been revised. Please refer to lines 56-61.

…Although, the benefits of SBJ training under handheld loading have been well established for mature athlete in man [11] and female [14, 15]. However, the majority of these studies did not include adolescent athletes. Therefore, a simple way that easy to training and more familiarization may be better for developmental stage athletes, using handhold load during SBJ training could have a number of interesting training implications…

Line 96: It would help to have a reference or a diagram of the modified Helen-Hayes configuration for study repeatability 

Response: Thanks for your comments. The diagram of the modified Helen-Hayes configuration please see figure 1.

Lines 153-160: How the subjects were instructed should be in the 2.2 procedure. How previous studies did it should be in Introduction.

Response: Thanks for your comments. The instruction of arm swing motion was in lines 76-78.

The discussion is not very convincing, especially with the unclear statement in the Introduction I mentioned above. By mentioning a lot of consistency with previous studies, it makes me feel even less novelty in this study. If the authors want to quantify the physical quality of the subjects, they should do a muscle strength test. The estimated torques from the experiments don’t tell much, since SBJ is a complicated exercise that requires a series of muscle cooperations. An increase in the joint moment won’t explain or prove muscle strength improvement. The authors also mentioned in lines 175-177 that reduced joint moments may improve jump performance. This further mixed the argument or statement the authors trying to make. Thus, I believe the authors need to spend more time clarifying their statement and distinguishing strength, explosiveness, performance, and technique, and how they should be quantified and related to each other in this study.

Response: Thanks for your comments. The paragraph was rephrased please refer to lines 177-195

The lower-extremity mechanical output exhibited improvement after training. Hip moment and power presented interesting results that the similar mechanical outputs in the Pre and PostL conditions, but were lower than the Post condition. Compare with Pre which explains the improvement of SBJ training on lower extremity mechanical output during 8 weeks. However, the smaller mechanical output under the PostL condition may be attributed to the increased pull under loading, which reduced the hip power [36]. Some studies have indicated that reduced joint moment may improve jump performance [37], which may due to arm swing effects [31,32] and found in this study which was reflected in the PostL condition, the handheld load reduced joint moment. Compare the results of Post and Pre conditions in present research, the higher joint moment in Post may prove the training effects that 8 weeks SBJ training was benefit for mechanical output. Improvements in lower extremity mechanical ability might be attributed to neuromuscular adaptations such as increased strength and explosiveness of the leg extensor muscles [6], and longer push-off phase may allowing the coordination of an optimal take-off position that slower knee and hip joint movement was benefit for the muscle activation with a load. It is possible that these neuromuscular and strength-explosive adaptations influenced the biomechanical factors related to jump actions, such as joint velocity of the take-off, which potentially contribute to produce a higher SBJ performance, cumulatively or individually. Overall, the results reflect the benefits of SBJ training under handheld loading.

Reviewer 2 Report

Dear authors: Congratulations on your work. I believe that the work would gain in clarity if, instead of a drawing, you attached photographic evidence of the tests carried out since, without a doubt, you must have the recordings made. Could you attach a photographic image of one of the tests carried out? In the methodology, they should indicate the markers that they used as references to carry out the measurements. Also indicate that the bibliography used is relatively old and only 2 articles are from the last 5 years. Can you justify this gap in knowledge?

I wish you luck

Author Response

Dear authors: Congratulations on your work. I believe that the work would gain in clarity if, instead of a drawing, you attached photographic evidence of the tests carried out since, without a doubt, you must have the recordings made. Could you attach a photographic image of one of the tests carried out? In the methodology, they should indicate the markers that they used as references to carry out the measurements. Also indicate that the bibliography used is relatively old and only 2 articles are from the last 5 years. Can you justify this gap in knowledge?

I wish you luck

Response: We appreciate your kind advice. The markers were added refer to Figure 1 & 2. And some references have been replaced. Please refer to references. Please refer to references:

  1. Scherrer, D; Barker, L.A.; Harry, J.R. Influence of Takeoff and Landing Displacement Strategies on Standing Long Jump Performance. Int J Strength Cond. 2022. 2(1).
  2. Tai, W.H.; Tang, R.H.; Huang, C.F.; Lo, S.L.; Sung, Y.C.; Peng, H.T. Acute Effects of Handheld Loading on Standing Broad Jump in Youth Athletes. Int. J. Environ. Res. Public Health 2021, 18, 5046. https://doi.org/10.3390/ijerph18095046.
  3. Zhou,H.; Yu,P.; Thirupathi, A.; Liang, M. How to Improve the Standing Long Jump Performance? A Mininarrative Review. A pplied Bionics and Biomech. 2020. 8829036.
  4. Gold, M.E.; Frisk, H.L.; Biggs, B.R.; Blankenship, M.J.; Ebben, W.P. Biomechanical analysis of loaded plyometric exercise, ISBS Proceedings 2021. Archive: Vol. 39: Iss. 1, Article 68.

Reviewer 3 Report

First of all, I want to congratulate the authors for the study. In fact, the article reports an important study for the academic and scientific community. Nevertheless, this article presents a set of situations that call into question the scientific and methodological criteria and rigor. Firstly, in the Introduction, several references are not recent (which does not mean that those presented by the authors are not important, but care must be taken in scientific writing and also use more current references). Regarding the Methodology, it is important to clarify a number of points: what type of study is it? Experimental? RCT? And guidelines that support this study? Regarding the participants, how is characterized the sample under study? In what year was the study carried out? Discussion/ Conclusion: What limitations does the study present? And what implications for practice has the study developed? What suggestions for future studies are there, in addition to those presented?

Author Response

First of all, I want to congratulate the authors for the study. In fact, the article reports an important study for the academic and scientific community. Nevertheless, this article presents a set of situations that call into question the scientific and methodological criteria and rigor. Firstly, in the Introduction, several references are not recent (which does not mean that those presented by the authors are not important, but care must be taken in scientific writing and also use more current references). 

Response: Thanks for your comments. Some references have been replaced. Please refer to references.

  1. Scherrer, D; Barker, L.A.; Harry, J.R. Influence of Takeoff and Landing Displacement Strategies on Standing Long Jump Performance. Int J Strength Cond. 2022. 2(1).
  2. Tai, W.H.; Tang, R.H.; Huang, C.F.; Lo, S.L.; Sung, Y.C.; Peng, H.T. Acute Effects of Handheld Loading on Standing Broad Jump in Youth Athletes. Int. J. Environ. Res. Public Health 2021, 18, 5046. https://doi.org/10.3390/ijerph18095046.
  3. Zhou,H.; Yu,P.; Thirupathi, A.; Liang, M. How to Improve the Standing Long Jump Performance? A Mininarrative Review. A pplied Bionics and Biomech. 2020. 8829036.
  4. Gold, M.E.; Frisk, H.L.; Biggs, B.R.; Blankenship, M.J.; Ebben, W.P. Biomechanical analysis of loaded plyometric exercise, ISBS Proceedings 2021. Archive: Vol. 39: Iss. 1, Article 68.

Regarding the Methodology, it is important to clarify a number of points: what type of study is it? Experimental? RCT? And guidelines that support this study? Regarding the participants, how is characterized the sample under study? In what year was the study carried out? 

Response: Thanks for your comments. The research is a randomized control trial, the guidelines was from Research Methods in Physical Activity (isbn:9789574966219), and the characterized of participants was fifteen adolescent male track-and-field athletes who were familiar with the SBJ participated in this study (mean age, body weight, height, and body mass index were 14.7 ± 0.9 years, 59.3 ± 8.0 kg, 1.73 ± 0.07 m, 19.8 ± 2, respectively). The manuscript was a part of an unpublished doctoral dissertation that was conducted from 2007 to 2009.

Discussion/ Conclusion: What limitations does the study present? And what implications for practice has the study developed? What suggestions for future studies are there, in addition to those presented?

Response: Thanks for your comments. The limitations have been revised. Please refer to lines 209-213. And the findings of this study serve as a reference for SBJ assessment and jump-related training. Most push-off-related variables demonstrated improvements after training, which indicated that the training effectively enhanced the explosive ability of the participants. Please refer to lines 217-219. future studies can assess whether the landing pattern of SBJ changes under handheld loading. In addition, to optimize training adaptations, SBJ training strategy should be adequately applied in a more complex, such as unilateral or different direction and to test an optimal loading before training that suitable for subjects. Please refer to lines 220-224.

Round 2

Reviewer 3 Report

After reading the second version, I consider the author introduced important aspects, and with answers to the questions of my opinion as reviewer. Thus, I am of opinion that the article has conditions to be published. Thank you for all.